



# Establishing a European Heliophysics Community (EHC)

Rumi Nakamura[a,b], Thierry Dudok de Wit[b,c], Geraint H. Jones[d], Matt G. G. T. Taylor[d], Nicolas André[e,f], Charlotte Goetz[g], Lina Z. Hadid[h], Laura A. Hayes[i], Heli Hietala[j], Caitríona M. Jackman[k], Larry Kepko[l], Aurélie Marchaudon[m], Adam Masters[n], Mathew Owens[o], Noora Partamies[p], Stefaan Poedts[q,r], Jonathan Rae[g], Yuri Shprits[s], Manuela Temmer[t], Daniel Verscharen[n], and Robert F. Wimmer-Schweingruber[u]

[a]Space Research Institute (IWF), Austrian Academy of Sciences (OEAW), 8042 Graz, Austria
[b]International Space Science Institute (ISSI), 3012 Bern, Switzerland
[c]LPC2E, OSUC, Univ Orleans, CNRS, CNES, 45071 Orleans, France
[d]ESA-ESTEC, Noordwijk, The Netherlands
[e]Institut Supérieur de l'Aéronautique et de l'Espace (ISAE-SUPAERO), Université de Toulouse, Toulouse, France
[f]Institut de Recherche en Astrophysique et Planétologie (IRAP), CNRS, Université de Toulouse, CNES, Toulouse, France
[g]Northumbria University, Newcastle-upon-Tyne, United Kingdom
[h]Laboratoire de Physique des Plasmas (LPP), CNRS, Observatoire de Paris, Sorbonne Université, Université Paris Saclay, Ecole polytechnique, Institut Polytechnique de Paris, 91120 Palaiseau, France
[i]Astronomy & Astrophysics Section, School of Cosmic Physics, Dublin Institute for Advanced Studies, Dunsink Observatory, Dublin D15 XR2R, Ireland
[j]Department of Physics and Astronomy, Queen Mary University of London, London E1 4NS, United Kingdom
[k]Astronomy & Astrophysics Section, School of Cosmic Physics, Dublin Institute for Advanced Studies Dunsink Observatory, Dublin, Ireland
[l]NASA Goddard Space Flight Center, Greenbelt, MD 20771, USA
[m]Institut de Recherche en Astrophysique et Planétologie (IRAP), CNRS, Université de Toulouse, CNES, 31028 Toulouse, France
[n]Mullard Space Science Laboratory, University College London, Dorking, United Kingdom
[o]Department of Meteorology, University of Reading, Reading RG6 6BB, United Kingdom
[p]Department of Arctic Geophysics, The University Centre in Svalbard, 9171 Longyearbyen, Norway
[q]Centre for Mathematical Plasma Astrophysics, Dept. of Mathematics, KU Leuven, 3001 Leuven, Belgium
[r]Institute of Physics, University of M. Curie-Skłodowska, Pl. M. Curie-Skłodowskiej 5, 20-031 Lublin, Poland
[s]GFZ Helmholtz Centre for Geosciences, 14473 Potsdam, Germany
[t]Institute of Physics, University of Graz, 8010 Graz, Austria
[u]Institute of Experimental & Applied Physics, Kiel University, 24118 Kiel, Germany

**Correspondence:** Rumi Nakamura (rumi.nakamura@oeaw.ac.at)

**Abstract.** Europe hosts a large and highly active community of scientists working in the broad domain of Heliophysics. This broad discipline addresses plasmas in the regions of space and atmosphere influenced by the Sun and solar wind. However, this community has historically been fragmented, both geographically and thematically, which has limited the potential for strategic coordination, collaboration, and growth. This has recently prompted a grass-roots community-building effort to foster

5 communication and interactions within the European Heliophysics Community (EHC). This white paper outlines the motivation, priorities, and initial steps towards establishing the EHC, and presents a vision for the future of Heliophysics in Europe. As a crucial first step of this endeavour, a dedicated EHC website is now available: https://www.heliophysics.eu/



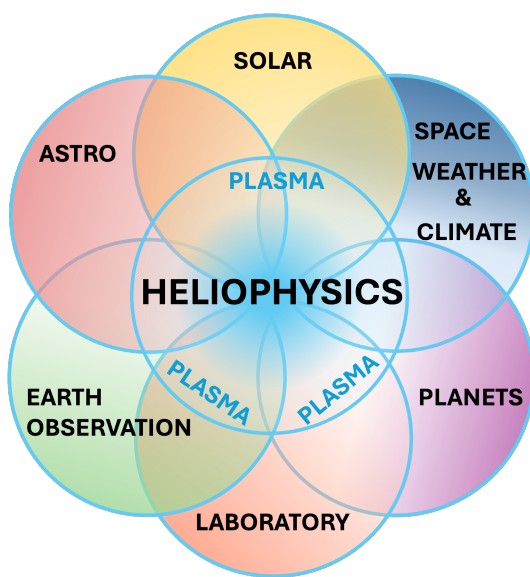

**Figure 1.** Illustration of the interconnected areas/topics of Heliophysics and neighbouring disciplines and communities, plasma and Heliophysics being denoted as crucial components of all surrounding disciplines.

## 1 Introduction

Heliophysics spans a wide range of disciplines covering the study of the Sun, its sphere of influence, and the different bodies in the Solar System and their interactions with the Sun. The field also covers a staggering wide range of scales, from the outer edge of the solar system to mesoscales, and down to the smallest (kinetic) scales at which electron dynamics determine plasma behaviour. Heliophysics is therefore inherently interdisciplinary, encompassing aspects of solar physics, space plasma physics, ionosphere-thermosphere physics, magnetospheric physics, planetary physics, small body physics, fundamental and applied space weather research, and more. Common to all these disciplines is the physics of (fully and partially) ionised plasmas, ranging from collisional to collisionless, and from magnetic to plasma energy dominated, see Fig. 1.

In Europe, a large and highly active community of scientists is working on these different aspects. However, this broad umbrella discipline has historically been fragmented geographically and thematically, which has made communication and collaboration cumbersome, even though fundamental plasma physics and techniques share a common thread. For this reason, there is a clear need for a European Heliophysics Community (EHC) that would improve communication (within the community and with other disciplines), foster collaboration and networking, drive innovation, and position Europe as a major player in addressing the scientific and practical challenges posed by our Sun. Such coordination would enhance Europe's leadership, long-term competitiveness, and resilience in Heliophysics research and its applications.

This community-building effort was ignited at the first *Heliophysics in Europe* Science Workshop, a week-long meeting held at the European Space Research and Technology Centre (ESTEC, Noordwijk) in October 2023. Motivated and inspired by the high interest and overwhelmingly positive community feedback received at a meeting held at the General Assembly of the



European Geosciences Union (EGU) in 2024, the second workshop, a comprehensive online edition, took place in November 2024. These discussions were complemented by a three-day forum hosted by the International Space Science Institute (ISSI, Bern) in January 2025. The authors of this paper are the participants of that forum.

This white paper synthesises the outcomes of these ongoing discussions and sets out a vision for building a joint community. It reflects contributions across the thematic scope of EGU's Solar-Terrestrial (ST) and Planetary and Solar System Sciences (PS), and also partly that of the Nonlinear Processes in Geosciences (NP) divisions. An EHC would facilitate the formation of interdisciplinary research teams, strengthen joint European Space Agency (ESA) mission proposals, and support more coordinated interdisciplinary research for cutting-edge science. A coordinated EHC would also promote interdisciplinary communication between early career and senior researchers and provide a platform to develop new ideas for space missions, ground-based facilities, modelling efforts, and shared software tools and data infrastructure across Heliophysics. Although this paper describes a European initiative, such an EHC is inherently international and aligns closely with the recent call for international cooperation in Heliophysics (Kepko et al., 2024).

After a brief historical overview in Sect. 2, we present several examples in Sect. 3 that illustrate the added scientific value of interdisciplinary interactions in Heliophysics. Sect. 4 concludes this work with suggestions and a vision of the way forward.

## 2 Background

For many years, European space scientists had expressed the desire to improve communication and interaction within the scientific community. In particular, members of the Earth's magnetosphere community considered organising meetings similar to those supported by the Geospace Environment Modelling (GEM) programme, an initiative of the USA's National Science Foundation (NSF) Division of Atmospheric Sciences, focusing on topical, interactive discussions instead of more formal, conference-style presentations. Around 2020, this discussion expanded to encompass a broader range of space plasma scientists under the Heliophysics umbrella, including those specialising in the Sun, planets and small bodies, as well as the ground-based and space weather communities.

A key issue in these discussions was that there was no single European entity representing this broad interdisciplinary field. For example, there was no equivalent to what is covered by the division on Solar-Terrestrial (ST), on Planetary and Solar-System Sciences (PS), and Nonlinear Processes in Geosciences (NP) in the European Geosciences Union (EGU). Clearly, the field's wide breadth and fragmentation also complicated interactions between sub-communities. This made responding to mission proposals and job or studentship calls challenging, as well as highlighting the logistical challenges of interacting across pre-existing sub-sections of the community, i.e. how to connect space- and ground-based observations. These discussions also raised the need for better support of shared tools and open-source software platforms, which support many cross-domain studies but lack coordinated development and long-term support. Within the Space Weather and Space Climate community, discussions about coordination has been ongoing for around a decade (Lilensten et al., 2021). These discussions ultimately resulted in establishing the European Space Weather and Space Climate Association (E-SWAN) in 2022, demonstrating the value of community action.



Driven by and in support of the European science community, the European Space Agency (ESA) has built a solid portfolio of Heliophysics missions. Missions that are of interest for Heliophysics commmunity such as Ulysses, SoHO, Cluster, Double Star, Solar Orbiter, Cassini-Huygens, Venus Express, Mars Express, Rosetta, BepiColombo, and Juice, all of which are the responsibility of the Science Directorate (D/SCI). However, in recent years, other ESA directorates have also developed and launched missions that address Heliophysics science. These include the Directorate of Earth Observation (D/EOP) with Swarm and other Earth Explorer missions (including the Earth Explorer 10 candidate, Daedalus, and follow-on activities carried out by the ESA-NASA Lower Thermosphere-Ionosphere Science Working Group (ENLOTIS, (Berthelier et al., 2024)) and Soil Moisture and Ocean Salinity mission (SMOS). The Directorate of Operations (D/OPS) has developed the Vigil mission, the Distributed Space Weather Sensor System (D3S), and the Space Weather Service Network. The Directorate of Human and Robotic Exploration (D/HRE) has developed many payloads, both for the International Space Station and for the Lunar Gateway, and the Directorate of Technology, Engineering, and Quality (D/TEC) has expertise in developing instrumentation and models for measuring and simulating environments throughout the heliosphere. The latter directorate is also responsible for the Proba-2 and Proba-3 missions, with support from D/SCI.

Recognizing the need to coordinate and communicate these activities more effectively across these directorates, the Director General of the ESA established the ESA Heliophysics Working Group (ESAHWG) in 2021. This cross-directorate group comprises representatives from the various directorates whose activities fall under the Heliophysics discipline. The intention is to improve internal interactions in this area within ESA. As part of their remit, the group was tasked with identifying synergistic activities and setting up community meetings to examine them. The ESAHWG organised a workshop entitled *Heliophysics in Europe*, supported by the community, to highlight cross-cutting activities in Europe and encourage discussion within the scientific community. Following this meeting, the community identified the need for structuring, establishing the EHC to identify cross-cutting topics, support early career colleagues, and develop a platform for long-term coordination. In parallel, there has been an international effort to recognise Heliophysics as a unified discipline with a substantial global community (e.g. Kepko et al., 2024). Therefore, the emergence of the EHC is well timed to build on this "heliophysical momentum" through grass-roots community action.

## 3 Examples of Heliophysics Science

Heliophysics covers a network of interconnected systems that are connected by a wide range of temporal and spatial scales, see in particular (Schrijver and Siscoe, 2009, 2010a, b; Schrijver et al., 2016, 2017, 2022). The Sun lies at the core, whose variability and solar wind interact with and shape the environment of all bodies within the Solar System through diverse processes such as heating, driving chemical reactions, sputtering of atmospheres and solid surfaces, space weathering (e.g. Hapke, 2001), ionisation, energizing plasma populations. These interactions shape magnetized bodies' magnetospheres, and create induced magnetospheres where mass-loading occurs. Defining the boundaries of Heliophysics has proven surprisingly difficult, many of our community discussions have revolved around this very question. The field is inherently interdisciplinary and resists a strict definition, drawing strength from its ability to connect diverse sub-disciplines and research approaches. It is therefore



clear that Heliophysics encompasses not only the study of the Sun and solar wind themselves, but also numerous subdisciplines in terrestrial and other planetary research areas, including, but not limited to, atmospheric science, magnetospheric physics, and cometary science. Another common thread of these studies is their multiscale approach (both temporal and spatial) in the sense that often, observing the interplay between processes operating at different scales is more important than observing the processes themselves. Due to the universality of the involved plasma processes, Heliophysics learns from and informs other fields, such as plasma-astrophysics and laboratory plasma physics (Koepke, 2008; Howes, 2018).

This Section presents selected examples of Heliophysics research to highlight its interdisciplinary nature. This list is not intended to be exhaustive; instead, it aims to demonstrate the vital importance of cross-disciplinary collaboration in achieving solid scientific progress.

## 3.1 Solar Physics and observing the source of Heliophysics Variability

Solar physics forms a central pillar of Heliophysics, providing the origin point for the magnetic and plasma structures that shape the heliosphere and drive variability throughout the solar system (Owens and Forsyth, 2013). From the solar dynamo and the emergence of magnetic fields to the heating of the corona and the eruption of flares and Coronal Mass Ejections (CMEs), solar physics addresses fundamental plasma phenomena with far-reaching heliospheric consequences.

A distinctive feature of solar physics within the context of Heliophysics is its dependence on remote sensing (e.g. Antonucci et al., 2020). Multi-wavelength observations, from EUV and X-ray to white light and radio, provide continuous imaging and spectroscopy of the solar atmosphere. These measurements are essential for diagnosing plasma temperatures, densities, and velocities, and for tracking the evolution of magnetic structures. Remote sensing enables the reconstruction of coronal magnetic fields, the detection of emerging flux, and the monitoring of flares and CMEs in real time.

Europe has established itself as a leader in solar physics through a combination of space missions and ground-based observatories. ESA-led missions such as SoHO and Solar Orbiter have provided critical insights into the solar atmosphere and solar wind (e.g. Velli et al., 2020). Ground-based observatories including the Swedish 1-m Solar Telescope, GREGOR telescope, Télescope Héliographique pour l'Étude du Magnétisme et des Instabilités Solaires (THEMIS), and radio facilities such as the Nancay Radioheliograph, and LOw Frequency ARray (LOFAR) contribute high-resolution imaging and radio diagnostics (e.g. Morosan et al., 2014). The European Solar Telescope (EST), currently under development, will further enhance Europe's capability to probe the structure and evolution of the solar magnetic field at small scales. ESA and European research teams have contributed to and benefited from international missions such as Hinode and IRIS, and are also involved in the upcoming Solar-C (EUVST) mission. These collaborations enhance scientific return and promote coordination across agencies and disciplines.

As such, solar physics is not an isolated discipline, but an integral part of Heliophysics. Its remote sensing capabilities, combined with theoretical modeling and data-driven approaches, provide the starting point for understanding the coupled Sun–heliosphere–planet system. In the next subsection, we explore how solar outputs, including the solar wind and eruptive events, structure the heliosphere and connect to space environments throughout the solar system.





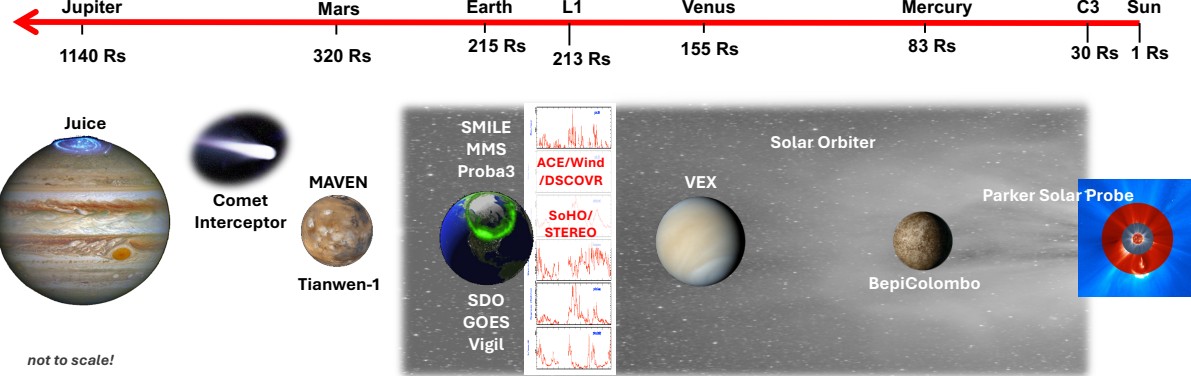

**Figure 2.** Selection of current, past, and future space missions that carry various types of instruments addressing heliophysics. These missions pursue a wide range of scientific objectives across different disciplines and over large distances, thereby deepening our understanding of the space plasma that fills our solar system. Remote sensing imaging data best cover the inner heliosphere, while in situ measurements provide local information about the plasma and magnetic field environment. Individual images taken from ESA/NASA. Figure adapted from Temmer (2021).

## 3.2 Connecting the Sun and Heliosphere

The Sun drives a supersonic solar wind flow, the origin of which is closely linked to one of the major open questions in contemporary physics: the nature of solar coronal heating (Cranmer and Winebarger, 2019). This solar wind inflates the heliosphere, a giant plasma bubble that surrounds the solar system and protects it from interstellar space. The topology and geometry of Earth's and other planets' magnetic fields are heavily influenced by the plasma stream, which mainly consists of protons and electrons emitted from the Sun's surface. Substantial variations in the solar wind's plasma and magnetic field characteristics stem from transient events, such as coronal mass ejections (CMEs) or co-rotating interaction regions (CIRs), as well as abrupt polarity changes due to the heliospheric current sheet. This plasma and the associated magnetic field constitute the external input to any magnetospheric processes. Hence, the different sources of the solar wind on the Sun structure the heliosphere (see e.g., Temmer, 2021; Zhang et al., 2021).

Our understanding of the heliosphere was shaped by early space missions such as the Pioneers (e.g. Fimmel et al., 1980) and Voyagers (Stone, 1977), by the twin Helios spacecraft (Porsche, 1977) and Ulysses (Wenzel et al., 1992), and today by a fleet of spacecraft including assets at the Sun-Earth Lagrange point L1: SoHO (Domingo et al., 1995), Wind (Harten and Clark, 1995), and ACE (Stone et al., 1998). In addition, there are STEREO (Kaiser et al., 2008) and Rosetta (Schwehm and Schulz, 1999; Glassmeier et al., 2007), while Parker Solar Probe (Fox et al., 2016), BepiColombo (Benkhoff et al., 2021), and Solar Orbiter (Müller et al., 2020) are today probing the inner heliosphere. After passing the heliospheric termination shock in 2004 (Stone et al., 2005) and the heliopause in 2013, Voyager 1 continues to return data from the local interstellar medium (Burlaga et al., 2022; Blinder, 2024). This fleet of missions allows us to track the solar wind from its origins to interplanetary space. Several of these missions are illustrated in Fig. 2.





Solar Orbiter is an excellent example of an interdisciplinary laboratory, addressing Heliophysics both in situ and from a re-
mote sensing perspective, its value augmented by collaboration with other missions. For instance, Telloni et al. (2023) exploited
simultaneous remote and local observations of the same coronal plasma volume, with Solar Orbiter/Metis and instruments on
Parker Solar Probe to determine the coronal heating rate in the slow solar wind, and tracked the radial evolution of turbulence
between Parker Solar Probe and Solar Orbiter (Telloni et al., 2021). Trotta et al. (2024) tracked the evolution of an interplane-
tary shock from 0.07 to 0.7 AU. Temmer et al. (2017) conducted a comprehensive study connecting solar flare–CME initiation
to their Earth impact by combining remote sensing, in situ measurements, and modeling, including 3D reconstruction and
magnetic flux rope analysis. Witasse et al. (2017) tracked a CME from 1 AU past Mars, comet 67P/Churyumov-Gerasimenko,
Saturn, and New Horizons en route to Pluto. The high-resolution images from Parker Solar Probe and Solar Orbiter also serve
as benchmark studies connecting small-scale structures related to solar wind interactions with evolving CMEs (Cappello et al.,
2024). Although not exhaustive, these examples illustrate the rich opportunities provided by the fleet of spacecraft in the solar
system and how they contribute to heliospheric science. Several European missions (Helios, Ulysses, SoHO, Rosetta, Bepi
Colombo, and Solar Orbiter) have played and are playing a key role in our understanding of the physics of the heliosphere.
They provide the critical link between observations at the smallest distances from the Sun (i.e. Parker Solar Probe) to the far-
thest reaches that space probes have ever reached (Voyager 1 and 2) and probe the interplanetary medium with unprecedented
resolution (e.g. Yang et al., 2023; Trotta et al., 2023). Recent studies show plenty of small-scale solar wind structures, such as
mini flux ropes and embedded magnetic fluctuations in the mesoscale regime, related to CMEs that might play an important
role in the interaction with the Earth's magnetic field (e.g. Lynch et al., 2023). Meso-scale solar wind structures may also
transfer energy from the Sun to geospace, see for example the review by (Viall et al., 2021).

In the context of solar physics, studying solar flares, CMEs, and particle acceleration has profound implications for a wide
range of areas within physics. It connects plasma physics, high-energy astrophysics and fundamental particle physics, estab-
lishing it as a vital foundation for interdisciplinary research. Recent results linking solar and magnetospheric physics emphasise
the various types of solar wind originating from different regions of the Sun and their impact on the magnetosheath (Koller
et al., 2024). This study revealed that classifications into quasi-parallel and quasi-perpendicular shocks are affected by the dif-
ferent types of solar wind types, suggesting importance of merging research on solar wind sources and dynamics with studies
on near Earth plasma environments such as shocks and the magnetosheath.

Among such solar wind sources interplanetary Coronal Mass Ejections (ICMEs) are large-scale solar eruptions that evolve
significantly as they travel through the heliosphere. Understanding their structure, expansion, and impact on space weather
requires multi-point observations. Fig. 3 shows an example of multiple spacecraft *in situ* observations of a CME at different
heliocentric distances (Davies et al., 2021) by Solar Orbiter, Wind, and BepiColombo that were closely aligned with a lon-
gitudinal separation of less than 5 °enabling to study radial evolution the ICME. Meanwhile STEREO-A remotely observed
the same event from a 75 °west of Earth, offering an optimal perspective to image the global structure of the CME. By com-
bining these remote-sensing data with *in situ* measurements, Davies et al. (2021) were able to track the large-scale shape of
the CME and its evolution through the inner heliosphere. The observations revealed a flattening of the flux rope cross-section,
suggesting that the ICME expansion was neither self-similar nor cylindrically symmetric. Additionally, a comparative analysis





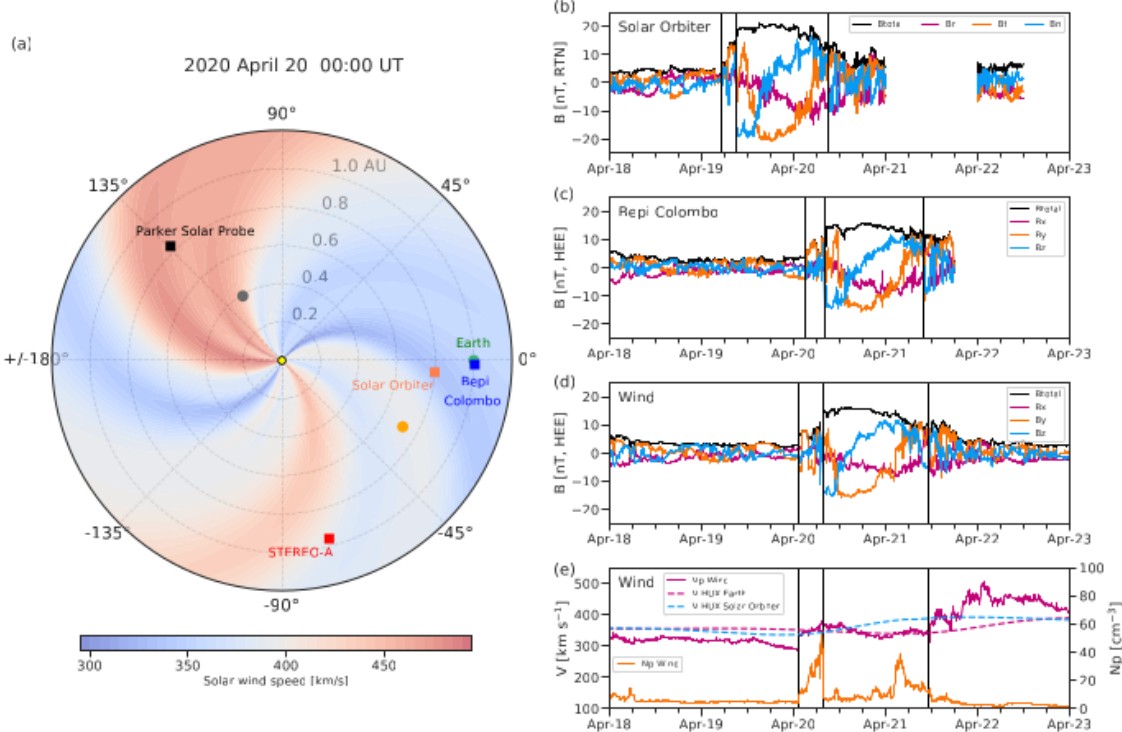

**Figure 3.** (a) Overview of spacecraft positions in the Heliosphere and *in situ* solar wind speed data up to 1 AU. (b) Solar Orbiter (c) BepiColombo, (d) Wind, magnetic field data, and (e) the wind proton speed and density (solid lines) and the WSA/HUX solar wind speed at Earth and Solar Orbiter (dashed lines). Vertical lines indicate, from left to right, the shock arrival time and the beginning and end times of the flux rope, determined visually. Reprinted from (Davies et al., 2021).

of the magnetic field strength between spacecraft indicated a deviation from the expected power-law dependence with distance
suggesting a complex evolution of the ICME as previously thought. Other examples of multi-point ICMEs studies can be found in (Weiss, A. J. et al., 2021; Palmerio et al., 2024)

This section would be incomplete without mentioning the imaging of the far side of the Sun, which is directly relevant to space weather forecasting (Heinemann et al., 2025). The techniques developed for this purpose draw on a variety of disciplines. The first observations were based on the illumination of hydrogen atoms in interplanetary space by intense Lyman $\alpha$ emissions
from active regions (Bertaux et al., 2000). Helioseismology is nowadays routinely used for forecasting purposes (Lindsey and Braun, 2017).




### 3.3 Magnetospheric Systems and Planetary Plasma Physics

Just as planetary geology is built on earlier scientific understanding of surface processes on Earth, planetary magnetospheric science has been strongly influenced by the earliest observations of the Earth's magnetosphere and the effect of the solar wind on it (Stern, 1996). Advancement from terrestrial to planetary magnetospheric studies has occurred much faster than in geology.

Decades of theoretical advances started to be confirmed when the first spacecraft, Explorer 1 exited the Earth's atmosphere and discovered the Van Allen Radiation Belts (Van Allen, 1958). Theoretical breakthroughs and experiments by luminaries such as Kristian Birkeland, who proposed that beams of electrons could create polar aurora and "polar magnetospheric storms" (Birkeland, 1908), now termed "substorms" (Akasofu and Chapman, 1963), and led to the idea that the magnetosphere-ionosphere could form a coupled system. Discoveries of fundamental plasma physical processes, such as magnetic reconnection led to the idea that planetary magnetospheres could really be coupled to the solar wind (Dungey, 1961), where sheared magnetic field lines could be reconfigured to thread a thin current sheet that separates two very disparate plasma regimes.

Fundamental scientific questions that arose from these revolutionary ideas, leading to missions to target the three-dimensional nature of near-Earth space (Cluster), what processes trigger polar magnetospheric storms (Time History of Events and Macroscale Interactions during Substorms; THEMIS), and what is involved in the process of magnetic reconnection (MMS). Over the last two decades, the study of the terrestrial magnetosphere has made major leaps forward thanks to multipoint measurements of these three missions, started first by the recently terminated Cluster mission. Cluster has pioneered our three-dimensional view of plasma physics at ion and fluid scales (Masson et al., 2024), including the 3-D nature of magnetic reconnection, of Kelvin-Helmholtz instability, and of fast flows and instability in stretched field lines of the magnetotail. The THEMIS (Angelopoulos, 2008) mission specifically targeted the scientific question of the causes and consequences of the polar magnetospheric storm a.k.a., substorm. THEMIS was designed to uniquely determine whether magnetic reconnection or plasma instability detonates the explosive energy release during a substorm (Angelopoulos et al., 2008; Rae et al., 2009). Both of these missions focused on ion and fluid scale physics to determine the global context and consequences of the physical processes. In contrast, the Magnetospheric Multi Scale (MMS) (Burch et al., 2016) mission focused entirely on microscopic kinetic/electron scales, respectively to understand in unprecedented detail how the fundamental plasma process such as magnetic reconnection occurs.

Taken in isolation, these are three missions that targeted specific scientific questions and gained unrivaled insight into each of these scientific questions. However, thanks to coordinated planning, and a large number of conjunction events between Cluster, THEMIS and MMS, their mutual benefit was far greater than their individual contributions, enabling different spatial and temporal scales to be probed simultaneously for the first time. These missions have gradually untangled the multiscale complexity of space plasmas, leading us to recognise that we are dealing with a system of systems, and one that should in future be planned for at the very start of the mission planning stage. Clearly, further international collaboration via "clusters of Clusters" is required to take the next big steps not only in magnetospheric science, but also in plasma physics across the universe (Retinò et al., 2022; Rae et al., 2022; Kepko et al., 2024).

The first significant planetary magnetic field observations beyond Earth were made at Jupiter, when Pioneer 10 passed through its magnetosphere in late 1973. This was closely followed by Mariner 10's first flyby of Mercury in March 1974.



These events occurred only around 15 years after Luna 1 first left Earth's magnetosphere. In the early years of the space age, terrestrial magnetospheric science taught us a great deal about how our planet's magnetosphere responds to changes in the heliospheric magnetic field and the dynamic and magnetic pressures of the highly variable solar wind.

The exploration of other planets that have a magnetosphere complements our knowledge and understanding of the terrestrial magnetosphere. The heliospheric environment changes with distance from the Sun. For example, Mercury's magnetosphere is much smaller than it would be if it were located at a greater heliocentric distance, where the heliospheric magnetic field is weaker and solar wind number density lower. In addition to the changing heliospheric environment at increasing distances from the Sun, the magnetised planets themselves differ from one another in planetary field strength, planetary rotation rate, alignment of their magnetic and spin axes and internal plasma sources. Thus the study of multiple planetary magnetospheres is

worth much more than the sum of the individual parts as it allows us to compare and contrast space plasma physics in a broad range of parameter spaces (Jackman et al., 2014).

Magnetospheres also provide a unique laboratory in the form of radiation belts, regions where highly energised electrons and various ion species are trapped. Several planets in the Solar System host these regions (Mauk and Fox, 2010), with the Earth's radiation belts being among the earliest discoveries of the fledgling field of space science in the late 1950s (Van Allen and

Frank, 1959). Given their dynamic nature, these belts have undergone significant scrutiny (Reeves et al., 2003; Thorne et al., 2007; Baker et al., 2004; Horne et al., 2005; Shprits et al., 2006). The causes of radiation belt changes are not well understood scientifically; operationally, these same particles can pose serious risks to satellites and astronauts (Baker et al., 1996) and have driven the growth of a key component of the applied discipline of Space Weather. Interest in the nature of other solar system radiation belts is growing, particularly in the case of Jupiter, where field strengths are 50 times greater than on Earth, leading

to a variety of complex physical processes. Understanding planetary radiation belts could provide a better understanding of exoplanetary systems and emissions (Roussos et al., 2022).

The Cassini-Huygens mission to Saturn is a prime example of an interdisciplinary project involving various instruments and scientific objectives. The mission's objectives included studying the planet's interior, atmosphere, magnetosphere, rings and moons. One of the Cassini mission's most famous discoveries was the plumes emanating from the icy moon Enceladus. This

was first observed through disturbances in magnetometer data (Dougherty et al., 2006). Subsequently, the plumes' appearance (Porco et al., 2006), composition and location near the south pole (Spencer et al., 2006) were confirmed using the other instruments on Cassini. Following the detection of magnetic field signatures during the first close flyby of Enceladus, the spacecraft's trajectory was altered to enable closer study of this intriguing moon. What began as a somewhat serendipitous event arguably became one of the mission's most significant discoveries, providing opportunities for instrument teams to

collaborate, and widening participation of science communities, e.g., scientists addressing habitability and origin of life in the Solar System.

The BepiColombo mission (Benkhoff et al., 2021) is a collaboration between ESA and the Japan Aerospace Exploration Agency (JAXA). It is another excellent example of an interdisciplinary mission that effectively integrates planetary, magneto-spheric, and solar wind science. Designed to study Mercury, the planet closest to the Sun, the BepiColombo mission comprises

two orbiters: the Mercury Planetary Orbiter (MPO) and the Mercury Magnetospheric Orbiter (MMO, also known as Mio).




Each orbiter is designed to investigate different yet interconnected aspects of the planet and its environment. By combining geological, geophysical, and chemical analyses of Mercury's surface with in-depth studies of its magnetosphere and its interactions with the solar wind, BepiColombo bridges multiple scientific disciplines. This approach allows scientists to investigate how the planet's thin exosphere is influenced by internal planetary processes and external solar activity, thereby advancing

our understanding of magnetospheric dynamics in an extreme solar environment. Thus, the mission exemplifies the synergy between different fields of space science, offering profound insights into Mercury's evolution and the broader workings of planetary systems. In addition to its primary objectives at Mercury, BepiColombo has already made substantial contributions to heliospheric science during its extended cruise phase. As discussed by Sánchez-Cano et al. (2025), the spacecraft has provided valuable observations of the solar wind, transient events, and planetary environments encountered during its flybys of

Earth, Venus, and Mercury. These measurements not only enhance our understanding of space weather in the inner heliosphere but also demonstrate the scientific value of planetary missions beyond their nominal operational phases.

The transport of mass, momentum and energy across boundaries or interaction between the particles and the electromagnetic fields are of interest not only in heliophysical plasmas, but also in plasmas all-over in the universe. Fundamental plasma processes such as magnetic reconnection, waves and turbulence in the boundary region have been studied throughout the Solar

System based on *in situ* and remote observations, as well as simulation studies. The solar system is the unique place in the universe where the fundamental plasma processes can be studied with *in situ* measurements for different plasma conditions from the diverse planetary and interplanetary environments. These unique observations allow validation of the models and compare with the remote observations of universal plasma processes.

An example of such plasma processes (shown in Figure 4) is the different studies of the Kelvin–Helmholtz instability in

various contexts: (a) near the Sun (Foullon et al., 2011) ; (b) in the solar wind at the boundary of a CME (Nykyri, 2024); (c) at the flank of the Earth's magnetopause (Hasegawa et al., 2004); (d) at the Mars dawnside ionopause (Wang et al., 2022); (e) at the Saturn dawnside magnetopause (Masters et al., 2010) (f) near the heliopause (Opher et al., 2003), and (g) at the edge of galactic jets (Walker et al., 2018). By comparing and contrasting these plasma processes, we can study how plasma flows interact with different obstacles throughout the Solar System and in an astrophysical context.

## 3.4   Investigating Coupled Ionosphere-Thermosphere-magnetosphere Systems

The coupled Ionosphere-Thermosphere-Magnetosphere (ITM) system is a highly dynamic, interconnected region of space close to magnetized planets such as Earth where solar and geomagnetic energy inputs drive complex physical processes, see Fig. 5. The magnetosphere, which is dominated by the planet's magnetic field, interacts with the solar wind, channelling energy and particles towards the high-latitude ionosphere. This energy input alters the ionospheric conductivity, driving currents

that influence thermospheric winds and temperatures through ion-neutral coupling. The ionisation of the primarily neutral thermosphere can also lead to chemical changes. All thermospheric effects can propagate throughout the ITM system and penetrate deep into the atmosphere. Additionally, the ITM is characterised by numerous interfaces, primarily the gradual transition from the neutral atmosphere (in the mesosphere and below) to the ionised atmosphere, and the shift from a collisional



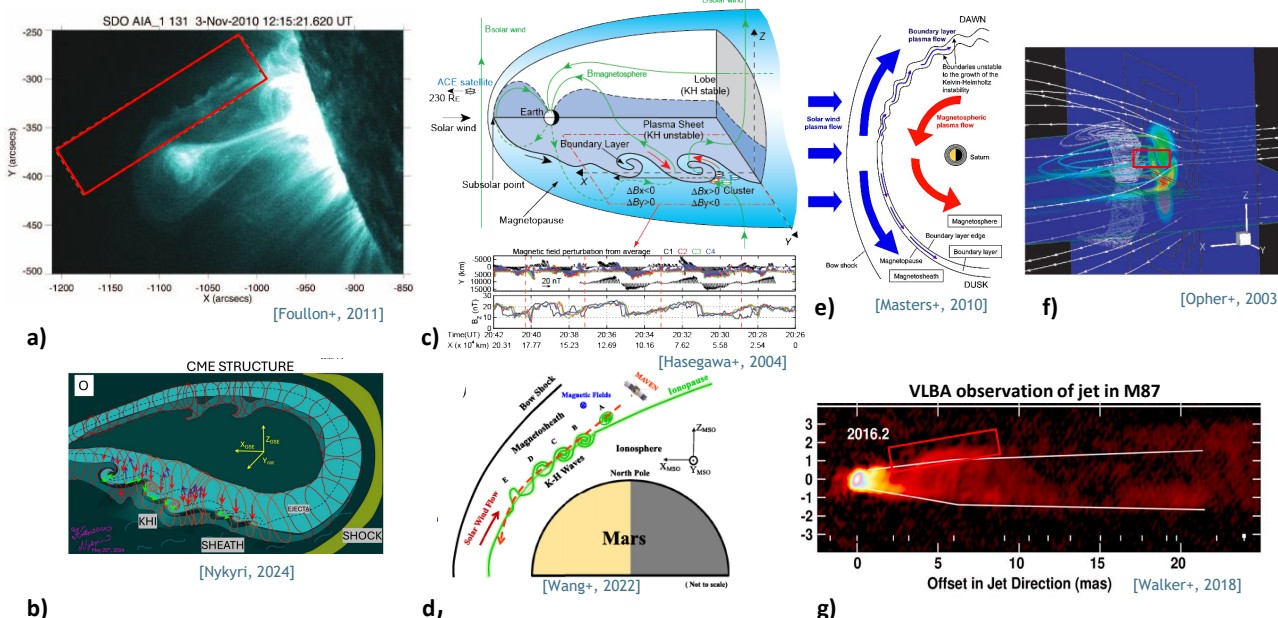

**Figure 4.** Kelvin-Helmholtz waves at (a) Sun (adapted from Foullon et al., 2011), (b) solar wind (adapted from Nykyri, 2024), (c) Earth (adapted from Hasegawa et al., 2004), (d) Mars (adapted from Wang et al., 2022), (e) Saturn (adapted from Masters et al., 2010), (f) heliopause (adapted from Opher et al., 2003), and (g) the edge of the galactic jets (adapted from Walker et al., 2018) .

to a collisionless regime. For these reasons, ITM science is inherently interdisciplinary. Therefore, it is crucial to understand
the ITM as a coupled system.

The atmosphere and the exosphere mark the end of the process chain for all solar system objects within the heliosphere. Here, the Sun's energy is deposited through processes such as Joule heating. Understanding the fundamental interactions in the ITM system is therefore essential, particularly on Earth, as these interactions can disrupt human infrastructures during intense space weather events.

The global electric circuit is another example of a system that links the space environment to the different layers of the atmosphere. This is exemplified by events such as sprites and elves during thunderstorms (Pasko et al., 2012), and more generally by the role of atmospheric electricity (Gordillo-Vázquez and Pérez-Invernón, 2021).

An additional degree of complexity arises from the coupling with the solar wind and magnetosphere at high latitudes. This coupling is responsible for the main deposition of energy in the system through ionospheric currents and particle precipitation,
which also cause the aurora. Excitation by solar illumination and thermospheric winds at the equator generates the equatorial electrojet and fountain. Such electrodynamic couplings are also observed on other planets, but they can differ greatly due to





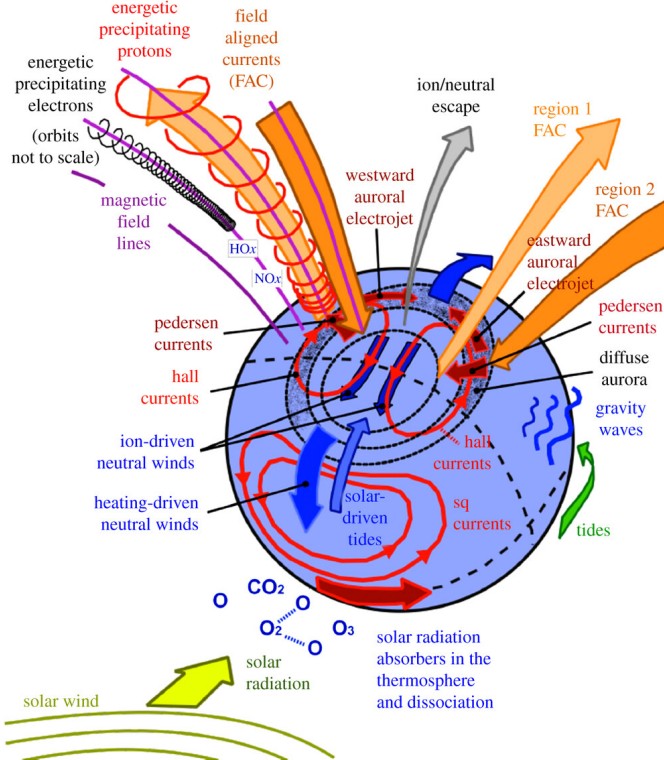

**Figure 5.** A schematic overview of ionosphere-thermosphere processes and their interaction with the magnetosphere and the solar wind (reprinted with authorisation from (Sarris, 2019)).

factors such as rapid planetary rotation, the presence or absence of an intrinsic magnetic field, and the density of the atmosphere. This is where past, present and future space exploration missions, combined with detailed physical models, can be crucial in enabling in-depth comparative planetology.

Luminous emissions, of which aurorae are the perfect example, illustrate the beauty of heliophysical processes. New discoveries are still being made in this field, such as Strong Thermal Emission Velocity Enhancement (STEVE) (MacDonald et al., 2018), fragments (Dreyer et al., 2021) and dynamic continuum emissions (Partamies et al., 2025). Aurorae have been observed on various planets in the Solar System across the electromagnetic spectrum, from radio waves to infrared, visible light and ultraviolet radiation. They can be regularly observed with space telescopes such as Hubble and JWST all the way to Uranus, 310     as well as with terrestrial radio telescopes, which also detect radio emissions from Jupiter's and Saturn's moons (e.g. Prangé et al., 2004).

The long-term development of empirical, physical and chemical models (e.g. Bruinsma, 2015; Laundal et al., 2022; Verronen et al., 2011) together with the existence of an extensive ground-based infrastructure (see Sec. 3.7) has established Europe as a leading provider of accurate descriptions of the electrodynamics, thermodynamics, and chemistry of the near-Earth space



environment. This enables us to unveil its complex physical and chemical processes. The coupling with the lower layers of the
atmosphere has also been explored in depth in recent years.

Last but not least, Europe has played a major role in missions such as Swarm (Friis-Christensen et al., 2006), which has
combined core magnetic field investigations with ionospheric connectivity and beyond, and SMILE (Branduardi-Raymont
et al., 2018), which investigates the coupling of the magnetosphere. By comparing inputs and impacts at the same time, and
associating ground based observations SMILE allow us to access physical parameters that cannot be measured directly, such
as conductivities and Joule heating.

By way of example, the physical ionosphere model called TRANSport au CARré (TRANSCAR) has been developed, main-
tained and updated over the last 30 years. Initially developed for high latitudes and open field lines (Blelly et al., 1996, 2005),
TRANSCAR was then extended by the IRAP Plasmasphere-Ionosphere Model (IPIM) to encompass closed field lines and the
description of the plasmaphere (Marchaudon and Blelly, 2015, 2020).

Inputs to TRANSCAR/IPIM can be derived from empirical models, directly from data, or adjusted from a combination of
data assimilated into these models. For example, the convection electric field can be derived from SuperDARN radar data
assimilated into dedicated empirical models (e.g. Thomas and Shepherd, 2018). Conversely, field-aligned currents can be
recovered from magnetometers on satellite missions such as CHAMP, IRIDIUM/AMPERE, and Swarm (e.g. Workayehu et al.,
2019; Pedersen et al., 2021). It is challenging to follow the empirical thermosphere model in dynamic situations, but it can
be fitted with density data derived from accelerometers on satellites such as CHAMP, Swarm, GRACE and GOCE. Once
optimised, TRANSCAR/IPIM simulations can be compared with other datasets for validation and interpretation, such as hmF2
and NmF2 data from ionosondes, electron density, electron and ion temperatures, and ion velocity from incoherent scatter
radars, as well as total electron content (TEC) maps from GNSS satellites. Pitout et al. (2015) simulated the impact of field
aligned currents observed with Swarm on the electrodynamics of auroral structures. Marchaudon et al. (2018) modelled the
effect of a solar high-speed stream (HSS) on the sharp depletion of ionospheric electron density in the F region. The model
results were successfully compared with observations from the EISCAT radar and/or Scandinavian ionosondes.

### 3.5  Weakly Magnetised Bodies

Venus, Mars, and Comets represent a special category of body in the solar system, in that they do not possess a global magnetic
field, and in the case of Venus and comets no intrinsic magnetic field at all. This makes their interaction with the solar wind
fundamentally different from that of Earth, Mercury, and the giant planets and presents a whole new laboratory of plasma
physics to explore. However, missions to these objects are always inter-disciplinary, and thus Heliophysics is often not a
priority in the mission objectives. This also offers opportunity to overlap and learn from other disciplines, e.g. about remanent
magnetic fields in the Martian crust or solar wind sputtering on surfaces. Even though the communities involved in these
objects are often small, they should not be overlooked when proposing new missions to funders.

For example, ESA's Rosetta mission (Glassmeier et al., 2007) observed the solar wind-comet interaction at comet 67P/Churyumov-
Gerasimenko for over two years as well as provided data from Earth, Mars and asteroid flybys. Rosetta has delivered data criti-
cal to our understanding of energy transfer in a two-component plasma and shown that aurora-like features can exist at comets





(Galand et al., 2020; Goetz et al., 2022). The main drawback of the Rosetta data is its lack of context for the solar wind and the lack of multi-point measurements. Comet Interceptor, ESA's new mission to a comet, has been designed to incorporate three spacecraft, which will provide, for the first time ever, three point measurements of the magnetic field in the plasma environment of a comet (Jones et al., 2024).

ESA's Venus Express has been instrumental in understanding the interaction of a completely unmagnetized planet with a strong solar wind (Svedhem et al., 2007). Venus Express was able to show that Venus' atmosphere is continuously eroded by the solar wind and found indications of magnetic reconnection in the tail as well as a plethora of common plasma waves in the entire magnetosphere (Futaana et al., 2017).

Mars is particularly interesting, as it does not possess a global magnetic field, but only local crustal magnetic fields. These induce asymmetries within the plasma environment that cannot be reproduced elsewhere (Vaisberg et al., 2018). ESA's Mars Express (Chicarro et al., 2004) has been measuring the plasma in the Martian magnetosphere and ionosphere since 2003. While making many fascinating discoveries on its own on atmospheric and magnetospheric dynamics (Martin et al., 2025) , it has also shown the strength of multi-point measurements through combination with NASA's MAVEN mission. For example, measurements by the Radiation Assessment Detector (RAD) (Hassler et al., 2012) on NASA's Mars Science Laboratory rover Curiosity (Grotzinger et al., 2012) are also contributing to our understanding of the propagation of CMEs and energetic particles through the solar system (see, e.g., Witasse et al., 2017; Kouloumvakos et al., 2024). These multi-point and multi-instrument measurements will be crucial for further understanding of plasma processes within the complex Martian magnetosphere (Sánchez-Cano et al., 2022).

Dusty plasmas have gradually become a discipline on their own, combining aspects of plasma physics, planetary science, astrophysics, materials science, and chemistry. Dusty plasmas are characterised by the presence of micro- to nanometer-sized dust grains that become electrically charged. Charged dust occurs in various regions: in the interplanetary medium (Horányi et al., 2009), in planetary rings, in cometary tails where solar UV radiation and the solar wind ionize gases and interact with dust (Price et al., 2019), in lunar and asteroid surfaces whose surfaces are eroded by the solar wind (Popel et al., 2018), and also in the Earth's mesosphere where noctilucent clouds are linked to dusty plasma produced by meteor ablation. In this context, the Moon stands out as an interesting laboratory not only for dusty plasmas but also for investigating kinetic processes in plasmas and the complex interactions between the solar wind and non-magnetised surface (Halekas et al., 2023). Some of these are illustrated in Fig. 6.

## 3.6 From Space to Laboratory Plasmas

As already mentioned, space plasmas offer the possibility to perform *in situ* investigations of fundamental plasma processes without significantly affecting the surrounding environment. Consequently, they frequently serve as natural laboratories for the observational study of plasma turbulence (Bruno and Carbone, 2013) and kinetic processes associated with shocks or reconnection (Verscharen et al., 2019).

Similarly, laboratory experiments on Earth aim to improve our understanding of related fundamental processes. Heliophysics thus provides many synergies with laboratory plasma experiments. Ground-based plasma devices such as the Large Plasma



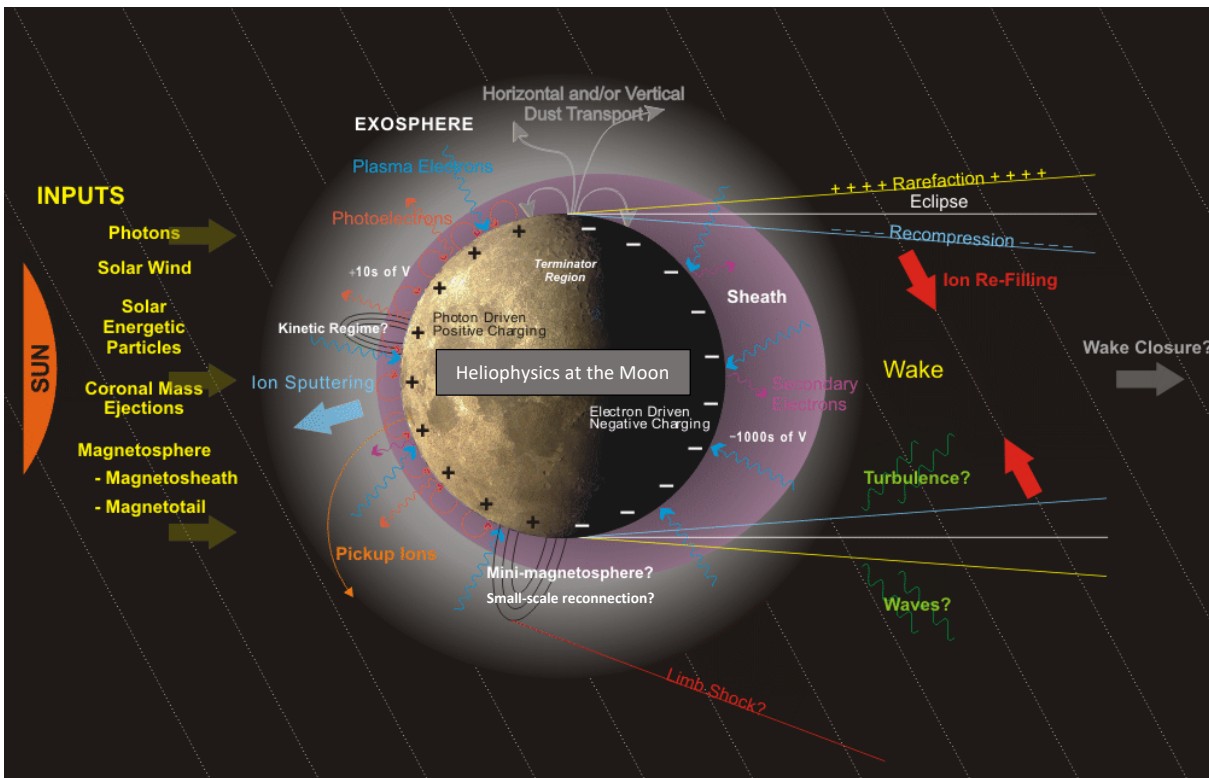

**Figure 6.** Physical processes acting in the lunar environment. Reprinted from (Halekas et al., 2023).

Device (LAPD), the Facility for Laboratory Reconnection Experiments (FLARE) and PHASe Space MApping (PHASMA) attempt to reproduce phenomena such as plasma waves, magnetic reconnection and particle acceleration (Gekelman et al., 385    1991; Ji et al., 2022; Shi et al., 2022). Unlike space plasmas, these devices provide reproducible and, to a certain extent, controllable conditions, thus complementing space observations of these processes when they occur sporadically. Comparisons between space and laboratory plasmas provide new insights into the fundamental physics of plasmas (Howes, 2018; Ji et al., 2023).

### 3.7   Observation Systems Beyond Space-Based: Ground-Based and Other Infrastructure

390    Historically, and particularly since the International Geophysical Year (1957–1958), Europe has been a driving force in the installation and operation of ground-based experiments in Heliophysics. This is one of the main reasons for the extensive coverage of instruments and the related scientific and technical expertise of European teams.

Thus, Europe plays a key role in the main instrument networks that provide information on the Ionosphere-Thermosphere-Magnetosphere (ITM) system. These include ground-based magnetometers (INTERMAGNET) for measuring ionospheric 395    currents, ionosondes (GIRO) and GNSS receivers (IGS) for determining ionospheric density and structure, and low-frequency





telescopes (LOFAR) and VLF receivers (AWDANet) for studying the dynamics of the mesosphere and thermosphere. Optical airglow observations and meteor measurements by optics and radars also support the studies of mesosphere and lower thermosphere. In the subarctic and Arctic regions, specific instruments dedicated to monitoring the polar cap and auroral regions and their complex electrodynamics have long been in place. These include incoherent scatter radars (EISCAT), coherent HF radars (SuperDARN), all-sky cameras for observing the aurora (e.g., MIRACLE and ALIS), riometers, Fabry–Pérot interferometers and scanning Doppler images. Some examples of the synergy between ground-based and space-based observations are given by Amm et al. (2005); Oberheide et al. (2015); Sarris (2019); Alfonsi et al. (2022).

The above mentioned ground-based observation networks offer complementary strengths to space missions and significantly enhance Earth system scientific understanding. While satellite instruments provide broad spatial coverage, ground-based networks offer high-resolution, localised measurements. Coordinated campaigns that align satellite overpasses with ground-based measurements with, for instance EISCAT and SuperDARN radars or optical networks with carefully designed special observation mode, allow for targeted investigations of dynamical ITM processes, such as substorms.

Europe also plays a key role in installing ground-based instruments in hard-to-reach regions (e.g. the subantarctic and Antarctic regions, sub-Saharan Africa, South America and East Asia) to monitor the equatorial region and southern auroral and polar zones. To this end, European teams are assisting local teams in the equatorial region with installing ground magnetometers to study the equatorial electrojet and GNSS receivers to investigate equatorial scintillations. They also train the regional teams to maintain and operate these instruments. Around the South Pole, Europe draws on its numerous national sub-Antarctic and Antarctic bases (in the UK, Italy, France, Norway, Sweden, Finland and Germany) to install instruments similar to those used in the Northern Hemisphere (e.g. SuperDARN radars, ground-based magnetometers, cosmic ray detectors and all-sky cameras) and to lead interhemispheric studies.

Several key European instruments, including large (>1 m) solar telescopes based in the Canary Islands, are also used for solar physics research. France, Germany and Sweden each own a telescope of this size, which they use to study the complex processes occurring on the Sun's surface and in filaments. The 4 m European Solar Telescope (EST), led by the Instituto de Astrofísica de Canarias (IAC, Spain), will also be deployed here.

Long-term solar radio observations have been performed by several countries, including France (Nançay radio observatory), Finland (Metsähovi radio observatory), Germany (Tremsdorf solar radio astronomy observatory) and Italy (Bologna and Cagliari sites), to track types II, III and IV, which are related to solar flares and CMEs (Pick and Vilmer, 2008). In addition, many smaller solar telescopes and coronagraphs are used across Europe, allowing for synoptic observations of the Sun's surface, corona and filaments. One of the main international networks is SAMNeT, which the UK leads. Ishii et al. (2025) provides a comprehensive overview of national and international instrumentation and networks that continuously observe the solar surface, Earth's magnetic field, and the ionosphere using both ground- and space-based instruments.

### 3.8 Long Time Scales: Space Climate

Long-term reconstructions of solar activity underpin a wide range of interdisciplinary science. Centennial- and millennial-scale solar activity reconstructions are used to estimate historical solar forcing, a key input to climate models and understanding



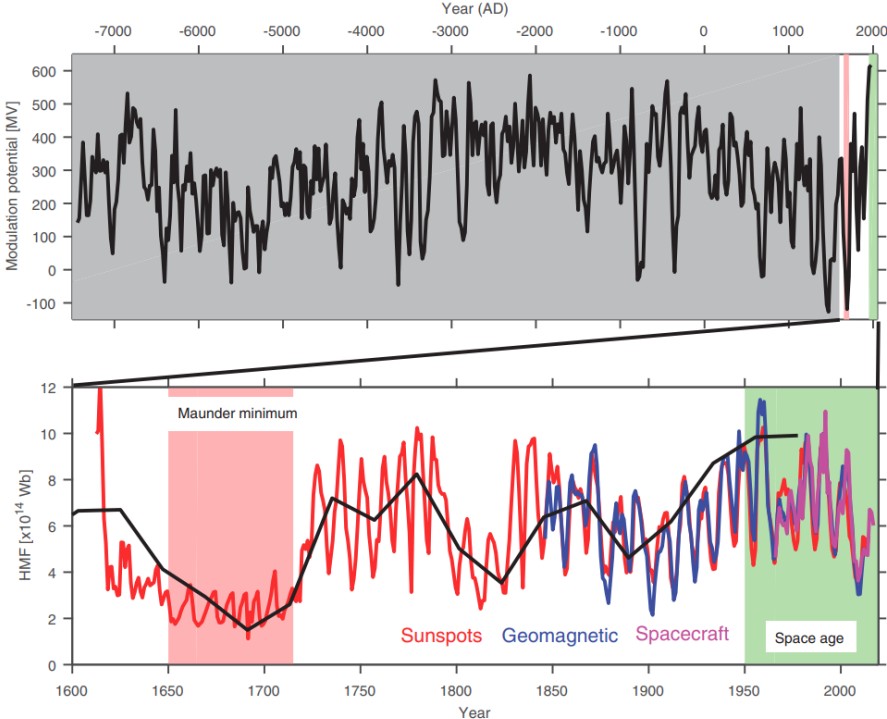

**Figure 7.** Reconstructions of solar activity from a range of sources, including direct spacecraft measurements, ground-based magnetometers, sunspot observers, and radionuclide information in tree rings and ice sheets. Reprinted from (Owens et al., 2018);

the terrestrial system in the past (Ermolli et al., 2013; Chatzistergos et al., 2023). Secular changes in solar activity, such as the Maunder minimum (1650-1715), also shed light on astrophysical observations of non-magnetically cycling sun-like stars (Baum et al., 2022). And more directly, understanding the range of solar activity that has occurred in the past provides the best estimate of what we can expect in the future, which is vital to ensuring societal and technological resilience to space weather (Owens et al., 2021).

435       The long-term solar activity reconstructions are themselves the product of interdisciplinary science. To extend reconstructions further into the past, it is necessary to use increasingly indirect solar activity proxies, often by calibrating against other (shorter duration, but more direct) measures or proxies. Direct, high temporal resolution spacecraft measurements of the near-Earth solar wind properties have been made near-continuously since the mid-1960s, and have been collated in the ongoing OMNI dataset (King and Papitashvili, 2005). Over a similar time scale, ground-based neutron monitors can reveal the rate at

which galactic cosmic rays (GCRs) produce nuclear reactions with atmospheric molecules (Usoskin et al., 2005). This, in turn, provides information about the solar magnetic field strength, which partially shields Earth from GCRs. Stretching back around 170 years, ground-based magnetometers measure the level of disturbance of the Earth's magnetic field resulting from the solar wind (Lockwood and Owens, 2011). These long time series of geomagnetic activity can then be coupled to atmosphere and





climate models to investigate the connections between solar effects on our past, current and future climate (Seppälä et al.,
445   2014).

Sunspot number counts, which measure visible solar activity, stretch back over four centuries. However, the construction of such a long-term dataset requires the interpretation of historical texts (Arlt and Vaquero, 2020; Clette et al., 2023). However, conversion to a more physical measure – such as open solar flux or total solar irradiance – requires considerable modelling and assumptions (Owens and Lockwood, 2012; Krivova et al., 2021). On similar timescales, records of auroral occurrence
provide about geomagnetic storms, but these data are complicated to use quantitatively, owing to changing patterns of human migration and hence sampling of geomagnetic latitude. But the longest reconstructions come from radionuclides produced in Earth's atmosphere by GCRs. These become locked up in tree rings and ice sheets, potentially enabling solar activity to be reconstructed back millennia (Muscheler et al., 2007; Brehm et al., 2021). This, however, requires extremely interdisciplinary science, spanning heliospheric, magnetospheric, high-energy physics, climate, glaciology, dendrochronology, etc.

**3.9   Beyond Space Plasmas of the Solar System**

Our list of examples of heliophysics, which highlights the interdisciplinary nature of research, would be incomplete without mentioning the outer edges of the heliosphere and their connection with the interstellar medium.

The heliosphere moves through the Very Local Interstellar Medium (VLISM) at a speed of approximately 26 km/s (Linsky et al., 2019). The interstellar plasma thus encounters an obstacle, the heliosphere, and must flow around it. Similarly, the
solar wind must at some point encounter the interstellar plasma. Because both the interstellar plasma and the solar wind are magnetized flows, they do not mix. The interstellar plasma must therefore flow around the heliosphere, and the solar wind must "wrap itself around" the heliosphere. The interface between the solar wind and the interstellar medium is called the heliopause. Information about the obstacle posed by the heliopause is transmitted back into the solar wind, where the termination shock forms, which deflects the solar wind and slows it down (Blum and Fahr, 1970). Voyager 1 passed the termination shock in
December 2004 at 94 au (Stone et al., 2005) (Voyager 2 passed it in August 2007 at 84 au). The region between the termination shock and the heliopause is called the heliosheath. Voyager 1 passed the heliopause and entered interstellar space in 2012 (Stone et al., 2013), Voyager 2 did so in 2018 (Stone et al., 2019). Recent observations by the Interstellar Boundary Explorer (IBEX, McComas et al., 2009) have shown that there is very likely no bow shock in the interstellar medium as had long been expected (McComas et al., 2012), however, this result is still being debated (Zank et al., 2013; Scherer and Fichtner, 2014;
Schwadron et al., 2015).

The neutral gas of interstellar matter enters the heliosphere uninhibited by the heliospheric boundary layers discussed in the previous paragraph where it can be ionized and picked up by the solar wind, forming a new population of suprathermal ions in the heliosphere, so-called pick-up ions (see, e.g.,  Kallenbach et al., 2000,  for a review). This mass loading of the solar wind slows it down by about 20% and heats it beyond about 20 - 30 au (Richardson and Stone, 2009). The pick-up process
results in a highly non-thermal velocity distribution function (Vasyliunas and Siscoe, 1976). This distribution is convected outward by the solar wind all the way to the termination shock, where it was widely believed to be further accelerated to form the anomalous component of cosmic rays (ACR, see, e.g.,  Jokipii, 1986). While the flux of low-energy particles increased as





Voyager 1 approached the termination shock (McDonald et al., 2003), it continued to increase beyond the termination shock, which indicates that ACRs are probably accelerated along the flanks of the heliosphere (McComas and Schwadron, 2006). An

alternative explanation involving magnetic reconnection at the compressed heliospheric current sheets around the termination shock was given by Drake et al. (2010).

Since its formation some 4.5 billion years ago, the solar system and heliosphere have revolved around the galaxy nearly 20 times and on its path must have experienced a wide range of interstellar environments. In high-density environments the heliosphere was compressed to within 25 au (Müller et al., 2009). Stronger compressions to within Saturn's orbit were probably

achieved by close-by supernova explosions in the galactic neighborhood (Wallner et al., 2020). Similarly to the records of short-term (on an astrophysical scale) variations in the heliosphere, there are archives of this "galactic voyage" that our heliosphere has undertaken during its history (McCracken et al., 2005; Scherer et al., 2006).

The heliosphere is the only "astrosphere" that we can investigate *in situ*. This system therefore serves as a model for understanding astrospheres surrounding stars that drive stellar winds (Weaver et al., 1977). These astrospheres strongly influence

the space environments of planet-hosting stars and have implications for astrobiology (Herbst et al., 2022). As such, the heliosphere provides a good example of how research in heliophysics can benefit astrophysics and vice versa, by drawing on the rich and diverse observations of remote astrospheres.

## 4 Conclusions and Vision

Heliophysics is an inherently interdisciplinary endeavour aimed at understanding our wider space environment. As such, it

brings together scientists from various subdisciplines who require different data sources and employ various analysis, theoretical and modelling approaches. For these activities to succeed, the scientists involved must engage with each other across disciplinary and methodological boundaries. This is a strategic goal of the EHC, which aims to bring fragmented communities together under a single, overarching theme. Here we briefly outline the key aspects and the recommended actions.

### 4.1 Low-threshold communication channels

To foster collaboration and innovation within the wider Heliophysics community, it is essential to establish communication platforms and provide a forum for exchanging ideas among scientists involved in Heliophysics research. Low threshold communication channels, such as a dedicated mailing list, a newsletter and a website, which are accessible and open to all community members, regardless of their career stage, are the backbone of this networking. The website is now active at https://www.heliophysics.eu/ and subscription to a regular newsletter is possible at https://spaceweather.gfz.de/helio-europe-mailing-list.

The latter includes a link for submitting announcements. Most recently the EHC has set up a linkedin account, for further communication and interaction (https://www.linkedin.com/company/heliophysics).





## 4.2 Exchange Experience Across Generations

The planning, development, and operation of space missions is a cross-generational effort. Therefore, the continuous training and carrying over of expertise is essential for the success of Heliophysics as a field. Heliophysics relies on well-trained Early
Career Researcher (ECR) who are motivated to pursue a scientific career in this field and to get involved in the cycle of space-mission development. In this regard, the support of ECR must lie at the heart of all activities by the ECR. As one component of this effort, we aim to establish a monthly EHC seminar series titled "HelioMeet". This initiative is designed to provide a platform for ECR to present their latest research, engage in meaningful discussions, explore interdisciplinary connections across various Heliophysics subfields, and encourage putting problems in a broader perspective. HelioMeet will amplify the
voices of early-career researchers within the broader scientific community, helping them establish professional networks, gain visibility, and contribute actively to shaping the future of Heliophysics.

## 4.3 Information Exchange for Different Data/Tools/Facilities/Disciplines

Our community already routinely practices openly sharing data from different sources and instruments. However, finding information about accessing data products can sometimes be difficult. Maintaining public data archives, communicating about
free access to data and making space-based and ground-based data linkable are thus key to the success of Heliophysics data analysis. Community efforts and a communication platform for effective information exchange are needed to leverage synergies between data providers. Heliophysics is connected to many other fields of science, such as exoplanet research, laboratory plasma physics and astrophysical plasma research. Therefore, the Heliophysics community is interested in and invested in exchanging with related fields outside its scope for mutual benefit.

## 4.4 Coordinated Data and Software Infrastructure for Heliophysics

Modern Heliophysics research increasingly depends on integrated data access, analysis, and modeling across traditionally separate domains, from solar observations to planetary magnetospheres, and combining multi-messenger data from both space- and ground-based platforms. However, the lack of shared infrastructure, standards, and sustained support for tools and data systems remains a major bottleneck to interdisciplinary science. This is being addressed with activities such as the International
Heliophysics Data Environment Alliance (IHDEA, Masson et al., 2024). To address Europe's scientific and strategic goals in Heliophysics, coordinated development investment in open-source software, data infrastructure, and community-led initiatives is essential.

A robust and interoperable ecosystem, where researchers can seamlessly work across missions, instruments, and domains, requires common data models, open documentation, reusable software libraries, and standardised metadata standards. Some
domains have developed powerful tools, particularly in Python, for example the community-led SunPy project (SunPy Community et al., 2020; Barnes et al., 2023), which provides core functionality for solar data analysis and is now foundational in the Heliophysics ecosystem. Yet many of these efforts across Heliophysics still operate in isolation, are inconsistently maintained, or lack interoperability, making it difficult to integrate data or replicate results across subfields. The Python in Heliophysics




Community (PyHC) (Barnum et al., 2023) builds on this foundation by promoting best practices, code compatibility, and sustainable development across a suite of open-source tools, including SunPy, and mission-specific packages. While PyHC originated in the US, there is a clear opportunity for European leadership, through ESA, national agencies, Horizon Europe programmes, and community networks, to shape its direction, and coordinate development across European tools and archives.

Europe already leads in open data access through resources like ESA's Heliophysics Archives. But infrastructure alone is not enough, long-term support for software maintenance, documentation, and developer coordination is essential, yet often underfunded or overlooked.

To enable interdisciplinary Heliophysics, data and software must be recognised as critical scientific outputs. Embracing open science and FAIR principles will support transparency, reuse, and broader participation. With targeted investment and coordination, Europe is well positioned to lead the development of a sustainable, interoperable ecosystem for Heliophysics research.

## 4.5 Outreach

The EHC encourages the exploitation of public events such as solar eclipses and Long Nights of Science to raise the profile of the community. In this respect, we aim to coordinate public events across Europe to promote public engagement. It is imperative to involve schools to ensure the next generation's interest in science. Citizen science involves inviting the general public to participate in active research by analysing data, finding events or attending public talks at local, regional, national or international levels. These activities can be coordinated by sharing presentation materials and data among EHC scientists through EHC repositories.

Finally, the EHC encourages the systematic use of the keywords "Heliophysics" and "EHC" (for example, as ORCID keywords) to increase the discipline's visibility and foster community cohesion. The term "Heliophysics" has now officially been added to the Merriam-Webster dictionary.

## 4.6 Scheme of Organisation

We envisage the EHC as a community-driven initiative where Heliophysics researchers are actively engaged at all career stages. Regardless of the organisational structure that the EHC adopts, all roles must be determined through transparent, open and inclusive practices. Based on our experience of national organisations such as MIST in the UK, ATST in France and AEF in Germany, we recognise that a low-overhead approach can lead to higher productivity, efficiency and inclusivity levels than a more formal organisational or management structure.

In an initial attempt to maintain momentum, the team of the ISSI Forum, joined by several volunteers, decided to set up four working groups that will focus on the following topics:

1. The EHC interim steering committee

2. Community engagement

3. Exchange experience across generations



4. Preparing the EHC workshop

As the authors of this paper do not represent the diversity of the EHC community, we would like to emphasise that the structuring phase is still ongoing. Depending on community interest and engagement, the working groups and their members will be revised and completed. The society is committed to equality, diversity, and inclusion, as will be reflected in the dedicated

section on its website and is planned to be addressed by a separate working group.

## 4.7 Communication and Coordination with other Organisations

There are national and international organisations that coordinate scientific communities engaged in Heliophysics. We believe that effective coordination and collaboration with these organisations is essential for our multidisciplinary research, which is based on internationally coordinated space and ground-based observations of the solar system. The Heliophysics research

pursued by EHC is directly relevant to space weather science, so close communication with E-SWAN is essential. However, while Heliophysics contains the science of space weather, it does not address more applied questions such as impacts on infrastructure (see also the introductory note in Schrijver et al., 2022).

The EHC can serve as a hub for different national organisations (MIST, ATST, AEF, etc.) to plan next-generation missions or observations at a European level, involving countries with and without such national organisations. We anticipate that this

coordination will enable EHC to join the worldwide ground observation network more effectively. EHC is anticipated to play a key role in the planning of the internationally coordinated space-based observations, such as the ISTP-NEXT proposal (Kepko et al., 2024), for solar and terrestrial missions, as well as other planetary missions throughout the solar system. Coordination with international programmes such as SCOSTEP (Scientific Committee on Solar-Terrestrial Physics) or ILWS (International Living with the Stars) that are promoting interdisciplinary research and collaboration will strengthen Heliophysics discipline

worldwide.



*Author contributions.* All authors contributed equally to the conceptualisation and the writing. The four lead authors coordinated the contributions.

*Competing interests.* One of the co-authors is a member of the editorial board of ANGEO

*Acknowledgements.* We gratefully acknowledge support by the International Space Science Insitute, ISSI, Bern, for supporting and hosting
the forum that resulted in this paper.



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
