# Peer review of "Establishing a European Heliophysics Community (EHC)"

_EGUsphere, 2025_

## Author Response (AR1)

**RC1**: 'Comment on egusphere-2025-3814', Tuija Pulkkinen, 29 Aug 2025  reply

*This paper is not a scientific contribution, but it reaches out to the European research community for better collaboration and coordination. As such, the paper is a valuable contribution, and my "no" answers above should not be held against publication.*

Thank you for carefully reading our manuscript. This paper is indeed not a conventional review paper, but is a white paper summarizing the important aspects for building an European Heliophysics community by addressing examples of interdisciplinary science topics of heliophysics.  We appreciate very much for your constructive comments.  We have addressed all your comments and made the following revisions. We hope the revised manuscript is acceptable for publication.

The changes in the text are given in trackchange.pdf file. We have revised Fig. 1 (adding proper credit) and Fig. 8 (modifying color scheme in response to the remarks in "review file validation") in addition to the new Fig. 2 and Fig 3. discussed below.

*The paper is very well written, I found only one place where the intent was not clear to me, this is the sentence starting on line 509 "Heliophysics relies..." I think I understand what the authors mean, but please check the language.*

Thank you for your positive comments, we have changed the sentence from "Heliophysics relies on well-trained Early Career Researcher (ECR) who are motivated to pursue a scientific career in this field and to get involved in the cycle of space mission development"  to "**The future of** Heliophysics relies on well-trained Early Career Researcher**s** (ECR**s**) who are motivated to pursue a scientific career in this field and to **advance the future** space mission development".

*On the presentation style, I have a few comments that the authors should consider, but I am by no means trying to force them. The first two sections are brief and to the point, stating the history and the need for the EHC. Section 3 is well-written and important, but it is also very long and I fear that it hides the vision section that comes at the end. You might consider having 1) Community efforts (your intro), 2) Gaining impact in organizations (your background), 3) Vision for the future, and 4) Science highlights.*

While we agree that the suggested structure by the reviewer is one possibility, we would like to highlight the science content EHC before the vision, because our vision (conclusions) are based on the science commonality and potential of interdisciplinary science discussed in section 3, together with the history and background discussed in section 1 and 2. Taking, however, also comments from the referee 2, we have added new diagrams and summary figure of the content of section 3 at the beginning of section 3. which we hope the reader can digest easier the main message of Section 3 and link to the Section 4.

*I feel that the science highlights as well as the vision would benefit from a summary diagram that would include all past, present and future missions, which form quite a fleet just reading through Section 3 (see the NASA corresponding one https://science.nasa.gov/heliophysics/missions/mission-fleet-diagram/). This could serve as part of your vision (what are the upcoming missions) as well as context for your highlights (what has been accomplished with the missions already past or in operation). I think that would also give the community an emotional sense of belonging, which might advance the realization of the vision.*

We have made a new Figure 2, a diagram of ESA-related recent and future missions similar to the NASA figure in section 2 Background in the paragraph when various missions from ESA directorate are named. The legacy missions are limited in the diagram not to make the figure too crowded, but are mentioned in the paragraph.  While the previous Figure 2 presented missions more tuned for the topic 3.2, the new Figure 2 covers also other areas as described in other subsections of section 3.   Furthermore, following the comments by the reviewer 2, we have added an illustration (Figure 3) of the research activities as EHC at the introduction part of Section 3. We think that these two figures should link the Section 3 content both to missions as well as to the interdisciplinary research topics.

*I know that the topic is challenging to address, but one glaring omission in section 4.7 is coordination and collaboration with other national space agencies with substantial resources. I leave it to the authors to decide whether to include a short mention how to keep track of what missions are planned globally, and how to prepare to take advantage of emerging opportunities for collaboration.*

Thank you for your important suggestion. While we have not explicitly mentioned the names of national agency, the sentence  "EHC is anticipated to play a key role in the planning of the internationally coordinated space-based observations, such as the ISTP-NEXT proposal (Kepko et al., 2024)"  are meant to indicate the collaboration among the different space agency.  To make our points more clear we have added the following sentence after it. "These communications among the different space agencies and the international scientific communities are expected to keep track of what missions are planned globally, and how to prepare to take advantage of emerging opportunities for collaboration."

**RC2**: 'Comment on egusphere-2025-3814', Sébastien Verkercke, 02 Sep 2025  reply

*While not a scientific contribution, this paper highlights the need for stronger collaboration within the European scientific community regarding heliophysics and is thus an important contribution for the scientific community. It clearly identifies the lack of a current European entity representing this broad field and emphasizes the interdisciplinary nature of heliophysics. It then proceeds to give a list of examples stressing the interdisciplinary aspects of the field, which covers the different scales, bodies, and structures encompassed by heliophysics, and reviews single- or multi-spacecraft, as well as ground-based observation techniques commonly used in the field. The paper concludes with the vision the authors have imagined for the European Heliophysics Community (EHC).*

Thank you for carefully reading our manuscript. This paper is indeed not a conventional review paper but is a white paper summarizing the important aspects for building an European Heliophysics community by addressing examples of interdisciplinary science topics of heliophysics.  We very much appreciate your constructive comments.  We have addressed all your comments and made the following revisions. We hope the revised manuscript is acceptable for publication.

The changes in the text are given in trackchange.pdf file. We have revised Fig. 1 (adding proper credit) and Fig. 8  (modifying color scheme in response to the remarks in "review file validation")  in addition to the new Fig. 2 and Fig 3. discussed below.

*The paper is well structured, and Section 3 provides a clear overview of the interconnections between the different fields, as shown in Fig. 1. However, some missions in Fig. 2 are not discussed in the text. An additional figure or diagram summarizing Section 3 could help the reader, as Section 3 is quite dense.*

Thank you for your important suggestion. Following also the suggestion of Reviewer 1, we have replaced Figure 2 with a more extended ESA-related recent and future mission diagram.  We place this figure in section 2 where it explains the background of ESA missions.  On the other hand, for section 3 we have added a new diagram describing the different discipline and area as a summary.  With these two diagrams, we consider that the relation between the missions and the various topics discussed in section 3 can be followed more clearly

*Section 4, which presents the vision for the EHC, seems somewhat hidden by the extent of Section 3. An additional diagram could be added to Section 4 to summarize the vision for the EHC and could be linked to the suggested diagram summarizing Section 3, as this would help the reader better understand the goals of the EHC in light of previous successes of interdisciplinary research in heliophysics.*

Since section 4 covers the different aspects to be consider for implementation of EHC, we think that the reader will find it easier to obtain the overview by simply reading the subtitles. We, however, made the relations between Section 3 and section 4 more clear by inserting following additional sentences.
1) We slightly modified the last sentence of Section 1 like:
   "Sect. 4 captures the main message of our work with a list of suggestions and a vision of the way forward."
2) We have added at the end of section 2 below sentences "In the next Section, we demonstrate this with a selection of examples that reveal the importance of multidisciplinarity in Heliophysics. However, the key message of this article and our vision for the future are detailed in Section 4."

*The paper is well written, but the sentence "Venus, Mars, and Comets, …" (l. 339) is misleading, as it could imply that these are the only bodies in this "special category".*

We have changed "Venus, Mars, and Comets represent a special category of body in the solar system, in that they do not possess a global magnetic field, and in the case of Venus and comets no intrinsic magnetic field at all." to
"A special category of body in the solar system are those that do not possess a global magnetic field, for example Venus, Mars, Comets and most Moons."

*Additionally, "Solar System" is sometimes written as "solar system", and some acronyms are defined multiple times (e.g., CME at lines 104 and 131).*

We have changed "solar system" to "Solar System" and deleted multiple definitions of acronyms for CME, EHC, ESA, MMS, THEMIS.